# Treatment Outcomes and Prognostic Factors of Gemcitabine Plus Nab-Paclitaxel as Second-Line Chemotherapy after Modified FOLFIRINOX in Unresectable Pancreatic Cancer

**DOI:** 10.3390/cancers15020358

**Published:** 2023-01-05

**Authors:** Takafumi Mie, Takashi Sasaki, Tsuyoshi Takeda, Takeshi Okamoto, Tsuyoshi Hamada, Takahiro Ishitsuka, Manabu Yamada, Hiroki Nakagawa, Takaaki Furukawa, Akiyoshi Kasuga, Masato Matsuyama, Masato Ozaka, Naoki Sasahira

**Affiliations:** Department of Hepato-Biliary-Pancreatic Medicine, Cancer Institute Hospital, Japanese Foundation for Cancer Research, Tokyo 135-8550, Japan

**Keywords:** pancreatic cancer, second-line treatment, gemcitabine plus nab-paclitaxel

## Abstract

**Simple Summary:**

There is no established standard second-line chemotherapy after modified FOLFIRINOX for unresectable pancreatic cancer. While gemcitabine plus nab-paclitaxel is often used as second-line chemotherapy after modified FOLFIRINOX, outcomes and prognostic factors of second-line gemcitabine plus nab-paclitaxel have been unclear. This study revealed that median overall survival (OS) and progression-free survival (PFS) of second-line gemcitabine plus nab-paclitaxel were 7.2 months and 3.6 months, respectively. Performance status, modified Glasgow prognostic score, and neutrophil-to-lymphocyte ratio were independently associated with overall survival. Our prognostic model using these parameters classifies patients into the good and poor prognosis groups. Median OS and PFS were longer in the good prognosis group than in the poor prognosis group. Efficacy of second-line gemcitabine plus nab-paclitaxel was notably limited, particularly in the poor prognosis group.

**Abstract:**

Outcomes and prognostic factors of second-line gemcitabine plus nab-paclitaxel (GnP) after modified FOLFIRINOX (mFFX) for unresectable pancreatic cancer were unclear. We retrospectively analyzed consecutive patients with unresectable pancreatic cancer treated with GnP after first-line mFFX treatment between March 2015 and March 2022 at our hospital. A total of 103 patients were included. Median overall survival (OS) from the start of first-line and second-line treatments was 14.9 months and 7.2 months, respectively. Median progression-free survival (PFS) was 3.6 months. Performance status, modified Glasgow prognostic score, and neutrophil-to-lymphocyte ratio were independently associated with OS. Our prognostic model using these parameters classifies patients into good (n = 70) and poor (n = 33) prognosis groups. Median OS and PFS were longer in the good prognosis group than in the poor prognosis group (OS: 9.3 vs. 3.8 months, *p* < 0.01; PFS: 4.1 vs. 2.3 months, *p* < 0.01). Grade 3/4 adverse events were observed in 70.9% of patients, with neutropenia being the most frequent. While GnP as second-line treatment was well-tolerated, efficacy of second-line gemcitabine plus nab-paclitaxel was notably limited, particularly in the poor prognosis group.

## 1. Introduction

Pancreatic cancer (PC) is unresectable at diagnosis in approximately 80% of cases, and its prognosis is poor [1]. Chemotherapy is widely used for the treatment of unresectable PC. Treatment outcomes of chemotherapy for unresectable PC have improved with the advent of FOLFIRINOX (FFX) and gemcitabine plus nab-paclitaxel (GnP) [2,3,4]. As superiority of one regimen over the other is still controversial, both FFX and GnP are positioned as standard first-line therapy [5].

There is no established standard of care for second-line chemotherapy in unresectable PC patients. After first-line GnP, the efficacy of nanoliposomal irinotecan plus fluorouracil and leucovorin and FFX as second-line chemotherapy has been reported [6,7,8]. After first-line FFX, gemcitabine-based therapy has been recommended as second-line treatment [9]. Since GnP has been reported to outperform gemcitabine monotherapy as first-line chemotherapy, GnP is often selected as second-line chemotherapy after FFX for patients in good general condition [3]. Most reports on GnP as second-line chemotherapy after FFX include small numbers of cases, and the prognostic factors of second-line GnP remain unclear [10,11,12,13,14,15,16,17,18]. Compared to FFX, a modified FOLFIRINOX (mFFX) regimen has fewer adverse events (AEs) and comparable efficacy, and mFFX has been widely used [19]. Therefore, a retrospective study was conducted to reveal outcomes and prognostic factors of second-line GnP after first-line mFFX.

## 2. Materials and Methods

### 2.1. Patients

We retrospectively analyzed consecutive patients diagnosed pathologically with unresectable PC, administered first-line mFFX treatment, and treated with GnP as second-line chemotherapy between March 2015 and March 2022 at our institution from a prospectively registered database.

### 2.2. Treatment

The regimen of GnP consisted of 125 mg/m^2^ nab-paclitaxel for 30 min, followed by 1000 mg/m^2^ gemcitabine for 30 min, on days 1, 8, and 15. Treatment cycles were repeated every 4 weeks. Prophylactic granulocyte colony stimulating factor was not used. In the event of AEs, dose reduction and/or delay were selected based on the discretion of the physician. The physician continued treatment until tumor progression, conversion surgery, intolerable AEs, patient refusal, or death. Third-line treatment was selected at the discretion of the physician.

### 2.3. Evaluations

Analyzed patient characteristics before second-line chemotherapy included age, sex, Eastern Cooperative Oncology Group performance status (PS), disease status (metastatic, locally advanced, or recurrent), tumor location (head or body/tail), metastatic site (liver, lung, distant lymph node, or peritoneum), and biliary drainage. Laboratory data including albumin, C-reactive protein (CRP), neutrophil-to-lymphocyte ratio (NLR) [20], carcinoembryonic antigen (CEA), and carbohydrate antigen 19-9 (CA19-9) were evaluated. Modified albumin and CRP were used to determine the modified Glasgow prognostic score (mGPS): patients with both hypoalbuminemia (albumin <3.5 g/dL) and elevated CRP level (>1.0 mg/dL) were scored as 2; patients with only elevated CRP level without hypoalbuminemia were scored as 1; and patients without elevated CRP level were scored as 0 [21].

The relative dose intensity (RDI) of gemcitabine and nab-paclitaxel was calculated as the actual dose intensity divided by the standard dose intensity for two courses after starting second-line treatment. Actual dose intensity was the actual dose divided by the actual period required to complete two courses of chemotherapy. Standard dose intensity was the standard total dose for two courses divided by eight weeks. Patients who did not complete two courses of chemotherapy were excluded from RDI calculations.

Follow-up computed tomography was performed every 8–12 weeks to assess efficacy of treatment. Treatment time and best response for mFFX as first-line chemotherapy were evaluated. CA19-9 was measured every month and CA19-9 reduction rate was evaluated by comparing the lowest CA19-9 level during second-line treatment with the CA19-9 level before the start of second-line treatment. The best responses for GnP as second-line chemotherapy were also evaluated. These analyses were performed in accordance with Response Evaluation Criteria in Solid Tumors, version 1.1 [22]. Categories and grades of AEs were evaluated according to Common Terminology Criteria for Adverse Events version 4.0 [23].

### 2.4. Statistical Analysis

Continuous variables are presented as medians (ranges) and were compared using the Mann–Whitney U test. Categorical variables are described as absolute numbers (proportions) and were analyzed using the χ^2^ test or Fisher’s exact test as appropriate. A *p* value < 0.05 was considered statistically significant. Progression-free survival (PFS) was measured from the first day of second-line chemotherapy to the time of disease progression, death, or last follow-up. Overall survival (OS) was measured from the first day of second-line chemotherapy to the time of death or last follow-up. Both PFS and OS were estimated using the Kaplan–Meier method and the Kaplan–Meier curves were compared by the log-rank test. To identify influencing factors for OS, the following variables were evaluated by univariate analysis: age (>70 years vs. ≤70 years), sex (male vs. female), PS (0 vs. 1/2), mGPS (0 vs. 1/2) and NLR (>3 vs. ≤3), CEA (>5 ng/mL vs. ≤5 ng/mL), CA19-9 (>37 U/mL vs. ≤37 U/mL), treatment duration of mFFX (≥4 months vs. <4 months), and best response of mFFX (complete response or partial response vs. stable disease or progressive disease). The Cox proportional hazards model was used to assess hazard ratios (HRs) and 95% confidence intervals (CIs) for OS. Factors in univariate analysis with *p* values < 0.05 were evaluated in multivariate analysis. The final analysis was conducted using follow-up data until 30 September 2022. All statistical analyses were performed with EZR ver. 1.40 (Saitama Medical Center, Jichi Medical University, Saitama, Japan) [24].

## 3. Results

### 3.1. Patient Characteristics

A total of 811 consecutive patients with unresectable PC received second-line chemotherapy between March 2015 and March 2022. Of these, 103 patients were treated with second-line GnP after mFFX. Patient characteristics are summarized in Table 1. Totally, 89 patients (86.4%) had metastatic PC. The most common metastatic site was the liver, observed in 60.2% of cases.

Table 2 shows the results of first-line mFFX. The median treatment period of first-line mFFX was 6.8 (range: 0.7–30.1) months. Overall response rate (ORR) and disease control rate (DCR) of first-line mFFX were 26.2% and 69.9%, respectively. Reasons for discontinuing first-line treatment were tumor progression (98.1%) and intolerance (1.9%). Grades 2 and 3 peripheral neuropathy were observed in 24.3% and 1.0% of patients, respectively, at the end of first-line mFFX.

### 3.2. Treatment Outcomes of Second-Line Chemotherapy

Ninety-one patients (88.3%) received two or more courses of second-line treatment. Median RDIs of gemcitabine and nab-paclitaxel were both 76.2% (range: 30.0–100%). The dose of gemcitabine or nab-paclitaxel was reduced in 83.5% (76/91) within two courses. The most common reason for dose reduction was hematologic adverse events, followed by fatigue and peripheral neuropathy.

The median follow-up period was 7.2 (range: 0.9–56.6) months. Median OS was 7.2 months (95% CI, 5.9–8.1 months) (Figure 1a). Median PFS was 3.6 months (95% CI, 3.0–4.4 months) (Figure 1b). The ORR and DCR were 7.8% and 63.1%, respectively (Table 3). CA19-9 reduction rate exceeded 50% in 40 patients (38.8%).

Median OS from the start of first-line mFFX was 14.9 months (12.7–16.9 months) (Figure 2).

### 3.3. Prognostic Factors

Univariate and multivariate analyses of factors affecting OS before the start of second-line chemotherapy are shown in Table 4. Multivariate analysis showed that PS = 0, mGPS = 0, and NLR ≤ 3 were independent prognostic factors. A prognostic index was created which assigns one point to each of the above variables. Patients with a score of 2 or 3 were included in the good prognosis group (n = 70), while those with a score of 0 or 1 were included in the poor prognosis group (n = 33). Median OS was longer in the good prognosis group (9.3 months; 95% CI, 7.4–10.6 months) than in the poor prognosis group (3.8 months; 95% CI, 3.1–4.2 months) (HR, 4.03; 95% CI, 2.56–6.34; *p* < 0.01) (Figure 3a). Median PFS was 4.1 months (95% CI, 3.6–5.3 months) in the good prognosis group and 2.3 months (95% CI, 1.8–2.9 months) in the poor prognosis group (HR, 2.11; 95% CI, 1.38–3.24; *p* < 0.01) (Figure 3b).

After the start of second-line GnP, CA19-9 reduction of more than 50% was significantly associated with OS (9.8 months with CA19-9 reduction of more than 50% vs. 5.6 months without CA19-9 reduction of more than 50%, *p* < 0.01). When comparing OS of patients with stable disease and progressive disease, significantly better OS was observed in the former (9.3 vs. 4.0 months, *p* < 0.01).

### 3.4. Adverse Events

Treatment-related AEs are shown in Table 5. No patient died of treatment-related causes. Grade 3/4 AEs were observed in 70.9% of the patients. Among grade 3/4 AEs, neutropenia occurred most frequently (46.6%). Peripheral neuropathy of all grades was observed in 86.4% of the patients, while grade 3/4 peripheral neuropathy was observed in only one patient.

### 3.5. Subsequent Treatment

Treatment after second-line chemotherapy was introduced to 51.0% of cases. The most common third-line treatment was S-1 (32.3%), followed by nanoliposomal irinotecan plus fluorouracil and leucovorin (6.3%). Two patients received conversion surgery, both of whom had liver metastases (Table 6).

## 4. Discussion

In this study, we evaluated the efficacy and AEs of GnP as second-line treatment after mFFX in a real-world setting. Second-line GnP achieved median OS of 7.2 months and median PFS of 3.6 months. Although the dose of gemcitabine and/or nab-paclitaxel was reduced in 83.5% of cases within two courses, GnP had an acceptable safety profile. Independent prognostic factors relating to OS were PS, mGPS, and NLR, which were used to classify the patients into the good and poor prognosis groups. Median OS and PFS were 9.3 months and 4.1 months in the good prognosis group, respectively, and 3.8 months and 2.3 months in the poor prognosis group, respectively. Efficacy of second-line GnP in the poor prognosis group was limited.

There is no established standard of second-line chemotherapy for unresectable PC after first-line mFFX. As gemcitabine-based therapy is recommended as second-line treatment after first-line mFFX [9], GnP is often selected for patients in good general condition, while gemcitabine monotherapy is selected for those in poor general condition. Median OS and PFS of second-line GnP have been reported to be 5.4–15.6 months and 2.8–5.8 months, respectively [10,11,12,13,14,15,16,17,18] (Table 7). The outcomes of this study were comparable to previous reports. On the other hand, median PFS of GnP as first-line treatment has been reported to be 5.0–8.4 months [3,5,25,26,27]. Outcomes of second-line GnP may be worse than those of first-line GnP because of poor general condition, high tumor volume, and resistance to chemotherapy. While ORR in this study (7.8%) was lower than that of previous reports, CA19-9 reduction exceeded 50% in 38.8% of patients. Although many patients did not achieve tumor shrinkage, the OS and PFS in this study were comparable to those in previous reports because a certain number of patients had more than a 50% reduction in CA19-9.

Prognostic factors after first-line treatment suggested by previous reports include PS, CEA level, CA19-9 level, PFS of first-line treatment, and a systemic inflammation-based prognostic score [7,28,29,30,31,32]. Multivariate analysis indicated that good PS, low mGPS, and low NLR were independent predictors of longer survival in this study. We created an original prognostic score by combining the prognostic factors identified in this study. Patients were classified into the good and poor prognosis groups, with a significant difference in median OS (9.3 months in good prognosis group vs. 3.8 months in poor prognosis group, *p* < 0.01). As median OS in the poor prognosis group was only 3.8 months, appropriate management including informed consent of end-of-life care should be given along with chemotherapy. Although a significant difference in PFS was also observed, the difference was less than two months (4.1 months in the good prognosis group vs. 2.3 months in the poor prognosis group, *p* < 0.01). It should be noted that components of our prognostic score may simply be associated with survival regardless of chemotherapy, instead of being predictors of the efficacy of second-line GnP.

Both mFFX and GnP are known to cause peripheral neuropathy. The frequency of grade 3/4 peripheral neuropathy of first-line mFFX and GnP was reported to be 1.8–9.0% and 2.8–17.0%, respectively [2,3,5]. The frequency of grade 3/4 peripheral neuropathy of second-line mFFX after GnP was reported to be 1.2–5.1% [8,33]. The reason for the low frequency of peripheral neuropathy in second-line mFFX might be due to appropriate dose reduction of nab-paclitaxel in first-line treatment and oxaliplatin in second-line treatment. In this study, grade 3 peripheral neuropathy at the start of second-line GnP was observed in only one patient, and no new patients developed grade 3/4 peripheral neuropathy during second-line GnP. The reason for the small number of patients with grade 3/4 peripheral neuropathy may be because 83.5% of patients had a reduced nab-paclitaxel dose and RDI of nab-paclitaxel was 76.2% within the first two courses of second-line GnP. Appropriate dose reduction before the development of grade 3/4 peripheral neuropathy may allow second-line GnP to be continued safely.

There are several limitations. First, this study was a retrospective cohort study conducted at a single center with no control group. Second, subsequent treatment was not standardized. Third, administration of therapeutic agents for peripheral neuropathy was not evaluated. Fourth, included patients had a mixture of metastatic, locally advanced, and recurrent PC.

## 5. Conclusions

In conclusion, GnP as second-line treatment was well-tolerated. Efficacy of second-line GnP was limited, particularly in the poor prognosis group.

## Figures and Tables

**Figure 1 cancers-15-00358-f001:**
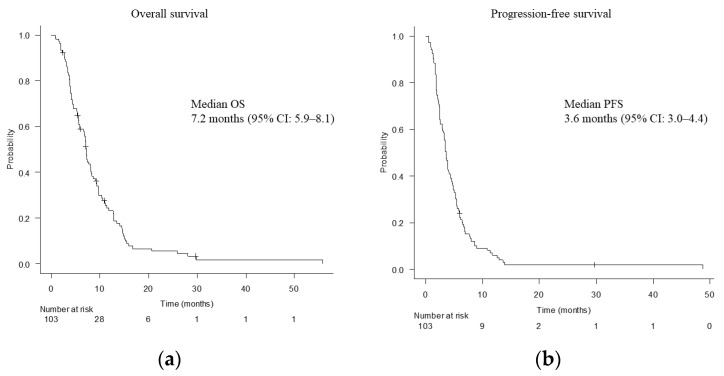
Overall survival (**a**) and progression-free survival (**b**). OS, overall survival; CI, confidence interval; PFS, progression-free survival.

**Figure 2 cancers-15-00358-f002:**
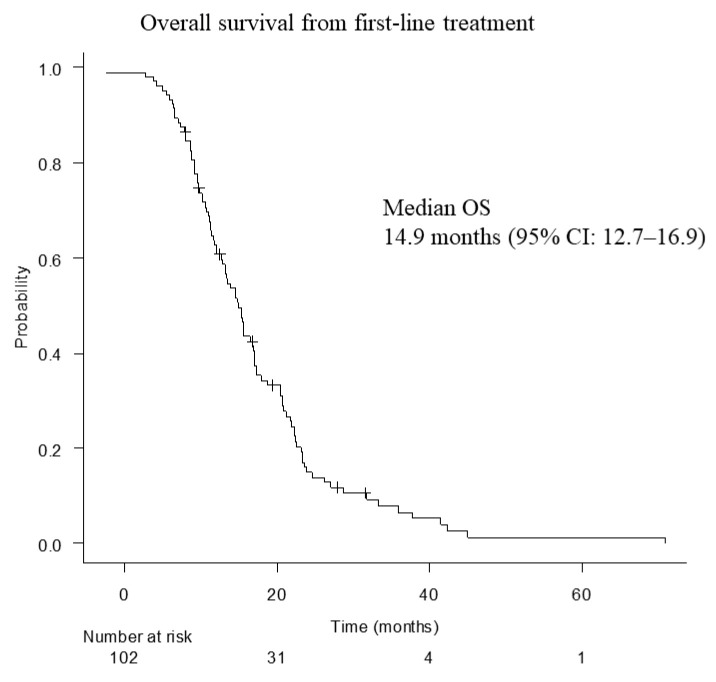
Overall survival from first-line treatment. OS, overall survival; CI, confidence interval.

**Figure 3 cancers-15-00358-f003:**
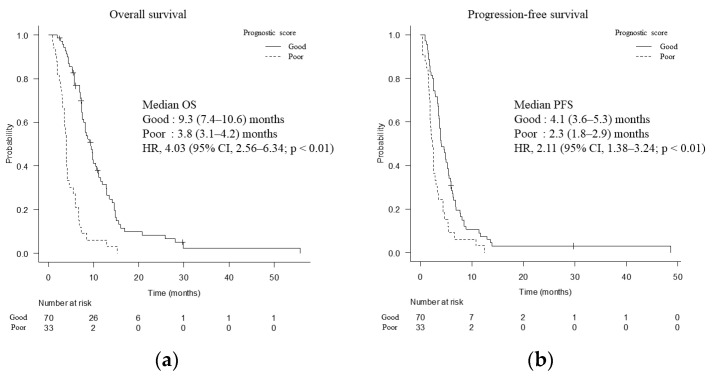
Overall survival (**a**) and progression-free survival (**b**) divided by prognostic factors. OS, overall survival; HR, hazard ratio; CI, confidence interval; PFS, progression-free survival.

**Table 1 cancers-15-00358-t001:** Patient characteristics.

		n = 103
Age (years)		
	Median (range)	67 (30–73)
Sex, n (%)		
	Male	58 (56.3)
	Female	45 (43.7)
PS, n (%)		
	0	72 (69.9)
	1	30 (29.1)
	2	1 (1.0)
Albumin (g/dL)		
	Median (range)	3.5 (2.3–4.4)
CRP (mg/dL)		
	Median (range)	0.41 (0.01–17.2)
CEA (ng/mL)		
	Median (range)	7.9 (0.9–4061.8)
CA19-9 (U/mL)		
	Median (range)	1069.3 (2.0–50,000)
mGPS, n (%)		
	Low (0)	64 (62.1)
	High (1, 2)	39 (37.9)
NLR, n (%)		
	≤3	60 (58.3)
	>3	43 (41.7)
Disease status (%)		
	Metastatic	89 (86.4)
	Locally advanced	9 (8.7)
	Recurrent	5 (4.9)
Tumor location (%)		
	Head	48 (49.0)
	Body and/or tail	50 (51.0)
Metastatic site, n (%)		
	Liver	62 (60.2)
	Lung	17 (16.5)
	Distant lymph node	28 (27.2)
	Peritoneum	35 (34.0)
Biliary drainage, n (%)		34 (33.0)

PS, performance status; CRP, C-reactive protein; CEA, carcinoembryonic antigen; CA19-9, carbohydrate antigen 19-9; mGPS, modified Glasgow prognostic score; NLR, neutrophil–lymphocyte ratio.

**Table 2 cancers-15-00358-t002:** Treatment outcomes and adverse events of first-line mFFX.

		n = 103
First-line treatment period (months)		
	Median (range)	6.8 (0.7–30.1)
Complete response, n (%)		0 (0)
Partial response, n (%)		27 (26.2)
Stable disease, n (%)		45 (43.7)
Progressive disease, n (%)		31 (30.1)
Overall response rate, %		26.2%
Disease control rate, %		69.9%
Reason for discontinuing first-line treatment, n (%)		
	Tumor progression	101 (98.1)
	Intolerance	2 (1.9)
Peripheral neuropathy at the end of first-line mFFX, n (%)		
	Grade 1	77 (74.7)
	Grade 2	25 (24.3)
	Grade 3	1 (1.0)

mFFX, modified FOLFIRINOX.

**Table 3 cancers-15-00358-t003:** Outcomes of second-line gemcitabine plus nab-paclitaxel.

		n = 103
Complete response, n (%)		0 (0)
Partial response, n (%)		8 (7.8)
Stable disease, n (%)		57 (55.3)
Progressive disease, n (%)		35 (34.0)
Not evaluable, n (%)		3 (2.9)
Overall response rate, %		7.8
Disease control rate, %		63.1
Completion rate of 2 courses, %		88.3
RDI, %, Median (range)		
	Gemcitabine	76.2 (30.0–100)
	nab-paclitaxel	76.2 (30.0–100)
Reason for reduction within 2 courses, n (%)		76 (83.5)
	Hematologic adverse event	47 (51.6)
	Fatigue	11 (12.1)
	Peripheral neuropathy	8 (8.8)
	AST/ALT increased	5 (5.5)
	Ascites	3 (3.3)
	Gastrointestinal symptoms	2 (2.2)

RDI, relative dose intensity; AST, aspartate aminotransferase; ALT, alanine aminotransferase.

**Table 4 cancers-15-00358-t004:** Factors influencing overall survival.

		Univariate		Multivariate	
		HR (95% CI)	*p* Value	HR (95% CI)	*p* Value
Age			0.40		
	>70	0.78 (0.44–1.39)			
Sex			0.35		
	Male	1.22 (0.81–1.84)			
PS			<0.01		<0.01
	0	0.44 (0.28–0.68)		0.44 (0.27–0.71)	
mGPS			<0.01		<0.01
	0	0.30 (0.19–0.47)		0.41 (0.26–0.66)	
NLR			<0.01		<0.01
	≤3	0.48 (0.32–0.73)		0.51 (0.32–0.80)	
CEA			0.12		
	≤5	0.71 (0.45–1.10)			
CA19-9			0.78		
	>37	1.07 (0.66–1.73)			
Treatment duration of mFFX			0.14		
	≥4 months	0.73 (0.48–1.11)			
Best response of mFFX			0.73		
	PR	0.92 (0.58–1.46)			

HR, hazard ratio; CI, confidence interval; PS, performance status; mGPS, modified Glasgow prognostic score; NLR, neutrophil-to-lymphocyte ratio; CEA, carcinoembryonic antigen; CA19-9, carbohydrate antigen 19-9; mFFX, modified FOLFIRINOX.

**Table 5 cancers-15-00358-t005:** Adverse events of gemcitabine plus nab-paclitaxel.

Adverse Events		n = 103	
		All Grades	Grade ≥ 3
Hematologic adverse events, n (%)			
	Anemia	102 (99.0)	30 (29.1)
	Neutropenia	80 (77.7)	48 (46.6)
	Thrombocytopenia	93 (90.3)	38 (36.9)
Non-hematologic adverse events, n (%)			
	Febrile neutropenia	1 (1.0)	1 (1.0)
	Nausea/vomiting	53 (51.5)	0 (0)
	Anorexia	15 (14.6)	0 (0)
	Diarrhea	41 (39.8)	0 (0)
	Constipation	65 (63.1)	0 (0)
	Stomatitis	21 (20.4)	0 (0)
	Alopecia	91 (88.3)	0 (0)
	Eruption	23 (22.3)	0 (0)
	Skin hyperpigmentation	7 (6.8)	0 (0)
	Fatigue	93 (90.3)	2 (1.9)
	Peripheral neuropathy	89 (86.4)	1 (1.0)
	Hypertension	3 (2.9)	0 (0)
	AST/ALT increased	91 (88.3)	6 (5.8)
	Creatinine increased	16 (15.5)	0 (0)
	Interstitial pneumonia	2 (1.9)	2 (1.9)

AST, aspartate aminotransferase; ALT, alanine aminotransferase.

**Table 6 cancers-15-00358-t006:** Subsequent treatment after second-line gemcitabine plus nab-paclitaxel.

	n = 49/96 (51.0%) *
S-1	31 (32.3%)
Nanoliposomal irinotecan plus fluorouracil and leucovorin	6 (6.3%)
Erlotinib plus gemcitabine	3 (3.1%)
S-1 with radiation	2 (2.1%)
mFFX	2 (2.1%)
Conversion surgery	2 (2.1%)
Gemcitabine plus S-1	1 (1.0%)
Clinical trial	1 (1.0%)
Folk medicine	1 (1.0%)
Second-line treatment continued after disease progression	2
Best supportive care	47
Transferred to another hospital	5

mFFX, modified FOLFIRINOX. * Patients continuing second-line treatment after disease progression or transferred to another hospital were excluded.

**Table 7 cancers-15-00358-t007:** Previous studies on second-line gemcitabine plus nab-paclitaxel.

Author	Year	Number of Patients	ORR (%)	DCR (%)	Median PFS (Months)	Median OS (Months)
Zhang Y et al. [10]	2015	28	18	46	NA	5.4
Portal A et al. [11]	2015	57	17.5	58	5.1	8.8
Dadi N et al. [12]	2017	47	NA	NA	2.8	7.5
Nguyen KT, et al. [13]	2017	30	NA	NA	3.7	12.4
Zhang H et al. [14]	2018	30	NA	NA	3.6	5.7
Mita N et al. [15]	2019	30	13.3	NA	3.8	7.6
Chae H et al. [16]	2020	102	8.5	73.6	4.6	9.8
Hayuka K et al. [17]	2021	25	12	96	5.3	15.6
Huh G et al. [18]	2021	40	15	87.5	5.8	9.9
Present study		103	7.8	63.1	3.6	7.2

OS, overall survival; PFS, progression-free survival; ORR, overall response rate; DCR, disease control rate; NA, not applicable.

## Data Availability

The datasets used and/or analyzed in the study are available from the corresponding author on reasonable request.

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
