# Peer review of "Treatment Outcomes and Prognostic Factors of Gemcitabine Plus Nab-Paclitaxel as Second-Line Chemotherapy after Modified FOLFIRINOX in Unresectable Pancreatic Cancer"

_cancers, 2023, doi:10.3390/cancers15020358_

Round 1

Reviewer 1 Report

Shimura M et al. retrospectively evaluated clinical outcomes and prognostic factors of second-line gemcitabine plus nab-paclitaxel (GnP) after modified FOLFIRINOX (mFFX) for 103 unresectable pancreatic cancer patients (mostly metastatic disease) from 2015-22 in a single institution. Overall survival (OS) and Median progression-free survival (PFS) was acceptable. Multivariate analysis revealed that performance status (PS), modified Glasgow prognostic score (mGPS) and neutrophil-to-lymphocyte ratio (NLR) were significantly independent prognostic factors.

They concluded that GnP as second-line treatment was well-tolerated, but efficacy of second-line GnP was notably limited, particularly in the poor prognostic group.

 Although this manuscript is well written and contains the interesting data, there were some comments and criticisms to be addressed.

Major comments.

1.       The authors collected the data of 103 UR-PC patients who received first-line mFFX treatment followed by GnP as second-line chemotherapy between March 2015 and March 2022. Please expose a total number of patients who received chemotherapy during the study period.

2.       How did the authors select 103 patients for mFFX followed by GnP? Clinical background in this study revealed relatively high proportion of young patients and metastatic disease but not locally advanced disease. Please explain this selection criteria for this regimen.

3.       In Table 4, absolute value of CA19-9 was included into univariate analysis. CA19-9 reduction exceeded 50% in 38.8% of patients. Did the authors evaluate association between OS or PFS and change in CA19-9 during the second-line chemotherapy? If it is statistically significant, it should be included in the multivariate analysis.

4.       Clinical response during chemotherapy may be consisted of anatomical, biological and conditional responses, which may be associated with better survival. In this study, conditional factors such as PS, mGPS and NLR were associated with improved survival. Did the authors find the difference in CA19-9 response between good and poor prognostic groups? In the other words, CA19-9 response during the second-line chemotherapy was related to the better survival?

I much appreciate for giving me an opportunity to review this manuscript.

Author Response

Dear Prof. Dr. Thomas Brunner

Special Issue Editor of Cancers

We have revised our manuscript entitled “Treatment Outcomes and Prognostic Factors of Gemcitabine Plus nab-Paclitaxel as Second-line Chemotherapy after Modified FOLFIRINOX in Unresectable Pancreatic Cancer” according to your kind recommendations. Our point-by-point replies to the reviewers’ comments are as provided below.

<Replies to Reviewer1’s Comments>

  1. The authors collected the data of 103 UR-PC patients who received first-line mFFX treatment followed by GnP as second-line chemotherapy between March 2015 and March 2022. Please expose a total number of patients who received chemotherapy during the study period.

Response: The manuscript has been revised as follows.

P3, L123-125

“A total of 811 consecutive patients with unresectable PC received second-line chemotherapy between March 2015 and March 2022. Of these, 103 patients were treated with second-line GnP after mFFX.”

  1. How did the authors select 103 patients for mFFX followed by GnP? Clinical background in this study revealed relatively high proportion of young patients and metastatic disease but not locally advanced disease. Please explain this selection criteria for this regimen.

Response: Since patients receiving mFFX for first-line chemotherapy were included in this study, many of them were relatively young patients. The patients with locally advanced PC might have metastatic lesions when first-line chemotherapy failed, so in the second-line chemotherapy, the proportion of locally advanced PC was lower.

  1. In Table 4, absolute value of CA19-9 was included into univariate analysis. CA19-9 reduction exceeded 50% in 38.8% of patients. Did the authors evaluate association between OS or PFS and change in CA19-9 during the second-line chemotherapy? If it is statistically significant, it should be included in the multivariate analysis.

Response: As you have indicated, CA19-9 reduction of more than 50% associated with OS. However, there is a problem with CA19-9 reduction is that it cannot be predicted before the start of second-line chemotherapy, so we don’t include CA19-9 reduction in the multivariate analysis in this study.

  1. Clinical response during chemotherapy may be consisted of anatomical, biological and conditional responses, which may be associated with better survival. In this study, conditional factors such as PS, mGPS and NLR were associated with improved survival. Did the authors find the difference in CA19-9 response between good and poor prognostic groups? In the other words, CA19-9 response during the second-line chemotherapy was related to the better survival?

Response: The rate of CA19-9 reduction of more than 50% in the good prognostic group and in the poor prognostic group were 44.3% vs 27.3%, respectively (p = 0.13). The OS of the patients with CA19-9 reduction of more than 50% was significantly better than those with CA19-9 reduction of less than 50% (9.8 months vs. 5.6 months, p < 0.01).

CA19-9 reduction of more than 50% is a prognostic factor after the start of treatment.

The manuscript has been revised as follows.

P7, L179-181

“After the start of second-line GnP, CA19-9 reduction of more than 50% was significantly associated with OS (9.8 months with CA19-9 reduction of more than 50% vs. 5.6 months without CA19-9 reduction of more than 50%, p < 0.01).”

We would like to express our sincere thanks to the reviewers for the constructive and positive comments.

Yours sincerely,

Corresponding author: Takashi Sasaki, MD, PhD,

Department of Hepato-Biliary-Pancreatic Medicine, Cancer Institute Hospital of Japanese Foundation for Cancer Research, 3-8-31, Ariake, Koto, Tokyo, 135-8550, Japan.

E-mail: sasakit-tky@umin.ac.jp

Phone: +81-3-3520-0111

Fax: +81-3-3520-0141

Reviewer 2 Report

This paper conducted a retrospective study (n=103) of pancreatic cancer patients treated with mFFX as the first-line treatment and with GnP as second-line chemotherapy in Japan population. Two survival outcomes were analyzed, showing a median OS of 7.2 months and a median PFS of 3.6 for the 2nd line GnP. A prognostic model based on performance status, modified Glasgow prognostic score, and neutrophil-to-lymphocyte ratio was able to predict OS and PFS.  Statistical analysis is reasonable, such as Kaplan-Meier method, log-rank test, and Cox proportional hazards model for survival analysis, χ2 test or Fisher’s exact test for categorical data, and Mann-Whitney U test for continuous variables. 

Two comments for consideration: (1) any survival difference between Stable disease vs. progressive disease, and (2) any adverse event associated survival outcomes.

Author Response

Dear Prof. Dr. Thomas Brunner

Special Issue Editor of Cancers

We have revised our manuscript entitled “Treatment Outcomes and Prognostic Factors of Gemcitabine Plus nab-Paclitaxel as Second-line Chemotherapy after Modified FOLFIRINOX in Unresectable Pancreatic Cancer” according to your kind recommendations. Our point-by-point replies to the reviewers’ comments are as provided below.

< Replies to Reviewer2’s Comments >

  1. Two comments for consideration: (1) any survival difference between Stable disease vs. progressive disease, and (2) any adverse event associated survival outcomes.

Response: (1) The OS was significantly better in the patients with stable disease than those with progressive disease (9.3 vs. 4.0 months, p < 0.01).

(2) No patient died of treatment-related causes. There was no difference in OS between patients who had Grade 3 or 4 AEs and those who did not have Grade 3 or 4 AEs (7.3 vs. 6.7 months, p = 0.63). Adverse events were acceptable, so second-line GnP could be safely administered.

The manuscript has been revised as follows.

P7, L181-183

“When comparing OS of patients with stable disease and progressive disease, significantly better OS was observed in the former (9.3 vs. 4.0 months, p < 0.01).”

We would like to express our sincere thanks to the reviewers for the constructive and positive comments.

Yours sincerely,

Corresponding author: Takashi Sasaki, MD, PhD,

Department of Hepato-Biliary-Pancreatic Medicine, Cancer Institute Hospital of Japanese Foundation for Cancer Research, 3-8-31, Ariake, Koto, Tokyo, 135-8550, Japan.

E-mail: sasakit-tky@umin.ac.jp

Phone: +81-3-3520-0111

Fax: +81-3-3520-0141

Round 2

Reviewer 1 Report

The authors appropriately responded to the Reviewers' comments.